# Phase II Trial of LINAC-Based STereotactic Arrhythmia Radioablation (STAR) for Paroxysmal Atrial Fibrillation in Elderly: Planning and Dosimetric Point of View

**DOI:** 10.3390/jpm13040596

**Published:** 2023-03-29

**Authors:** Ilaria Bonaparte, Fabiana Gregucci, Antonio Di Monaco, Federica Troisi, Alessia Surgo, Elena Ludovico, Roberta Carbonara, Eleonora Paulicelli, Giuseppe Sanfrancesco, Christian De Pascali, Nicola Vitulano, Federico Quadrini, Maria Paola Ciliberti, Imma Romanazzi, Fiorella Cristina Di Guglielmo, Davide Cusumano, Roberto Calbi, Massimo Grimaldi, Alba Fiorentino

**Affiliations:** 1Department of Radiation Oncology, General Regional Hospital “F. Miulli”, 70021 Acquaviva delle Fonti, Italy; 2Department of Cardiology, General Regional Hospital “F. Miulli”, 70021 Acquaviva delle Fonti, Italy; 3Department of Clinical and Experimental Medicine, University of Foggia, 71122 Foggia, Italy; 4Department of Radiology, General Regional Hospital “F. Miulli”, 70021 Acquaviva delle Fonti, Italy; 5Department of Cardiology, Policlinico “G. Rodolico”-Azienda O.U. Policlinico “G. Rodolico”, 95123 San Marco Catania, Italy; 6Department of Radiation Oncology, Mater Olbia Hospital, 07026 Olbia, Italy; 7Department of Medicine, LUM Giuseppe Degennaro University, 70010 Casamassima, Italy

**Keywords:** radiosurgery, atrial fibrillation, stereotactic radiotherapy, arrhythmia

## Abstract

**Purpose:** Approaching treatment for elderly patients with atrial fibrillation is difficult. A prospective phase II trial evaluating LINAC-based stereotactic arrhythmia radioablation (STAR) safety in this population started in 2021. Dosimetric and planning data were reported. **Materials and Methods:** A vac-lock bag was used for immobilization in the supine position and a computed tomography (CT, 1 mm) was performed. The clinical target volume (CTV) was defined as the area around the pulmonary veins. An internal target volume (ITV) was added to the CTV to compensate heart and respiratory movement. The planning target volume (PTV) was defined by adding 0–3 mm to the ITV. STAR was performed during free-breathing with a PTV prescription total dose (Dp) of 25 Gy/1 fraction. Flattening filter-free volumetric-modulated arc therapy plans were generated, optimized, and delivered by TrueBeam^TM^. Image-guided radiotherapy with cone-beam CT and surface-guided radiotherapy with Align-RT (Vision RT) were employed. **Results:** From May 2021 to March 2022, 10 elderly patients were treated. Mean CTVs, ITVs, and PTVs were 23.6 cc, 44.32 cc, and 62.9 cc, respectively; the mean prescription isodose level and D2% were 76.5% and 31.2 Gy, respectively. The average heart and left anterior descending artery (LAD) Dmean were 3.9 and 6.3 Gy, respectively; the mean Dmax for LAD, spinal cord, left and right bronchus, and esophagus were 11.2, 7.5, 14.3, 12.4, and 13.6 Gy, respectively. The overall treatment time (OTT) was 3 min. **Conclusions:** The data showed an optimal target coverage, sparing surrounding tissue, in 3 min of OTT. LINAC-based STAR for AF could represent a valid non-invasive alternative for elderly patients who were excluded from catheter ablation.

## 1. Introduction

More than 40 million individuals in the world are affected by atrial fibrillation (AF) and, age being a prominent risk factor, such incidence is expected to grow, considering the increasing average lifespan. The current international guidelines recommend pulmonary vein (PV) isolation with catheter ablation (CA) in symptomatic patients, refractory to antiarrhythmic therapy (AAT) [1].

In elderly patients, which represent the majority of the AF population, the treatment approach is quite complex and invasive: paroxysmal AF is difficult to treat with drugs, since it can cause an alternation between alternate sinus bradycardia and fast-rate AF. As regards the CA approach, the high incidence of complication makes the study of more conservative methods a reasonable consideration [2].

One of the most promising and novel alternative approaches for the management of such patients is stereotactic radiotherapy, a safe and effective discipline that uses a high radiation dose to produce relevant cell damage through multifactorial processes (DNA double-strand breaks, apoptosis, vascular damage, and ischemic cell-death).

In recent years, several stereotactic arrhythmia radioablation (STAR) clinical approaches have been published for ventricular tachycardia, while only studies on animals and three case reports with CyberKnife have been reported for AF [3,4,5].

Animal models for radioablation to block signals at the pulmonary venin antrum showed that scarring effects arise with dosages over 30 Gy [6,7,8]. Recent studies introduced the concept of radio modulation in the STAR treatment, according to which 25 Gy single-fraction radiation does not increase cardiac fibrosis, but it modulates the expression of sodium channels, producing a marked decrease in the arrhythmic burden [9,10].

Based on this background, a prospective phase II trial was designed, intended to evaluate the safety of LINAC-based STAR (ClinicalTrials.gov: NCT04575662) in elderly patients, with the first clinical data of five patients recently published [11].

The aim of this study was to report the dosimetric and planning data of the first 10 enrolled patients, proposing a novel planning strategy for the physical management of this treatment.

## 2. Methods

The NCT04575662 Phase II study was approved by the local Ethics Committee and all patients signed informed consent forms. As previously reported [11], the following inclusion criteria were considered: age higher than 70 years; symptomatic paroxysmal AF; intolerance or non-response to AAT.

The primary endpoint was the 1-month post-STAR safety check, in terms of complete STAR delivery with no acute treatment-related adverse events above G3, assessed according to the Common Terminology Criteria for Adverse Events (version 5.0).

Secondary endpoints were reductions in AF episodes, reduction in AAT, and overall survival.

The sample size was planned for 20 cases based on 95% success for the primary endpoint, with a significance level of 5% and a statistical power of 90%.

A vac-lock bag was used for patient immobilization in the supine position. Three computed tomography (CT) scans were acquired: (1) free-breathing CT for dose calculation; (2) 4-Dimension CT (4D-CT) for motion evaluation with 10 phases; and (3) CT with contrast for anatomical accuracy [11]. The following organs at risk (OaRs) were delineated referring to the international atlas [12,13,14]: lungs, trachea, main bronchus, esophagus, breasts, aorta, superior and inferior vena cava, pulmonary artery, pulmonary veins, left atrium, right atrium, left ventricle, right ventricle, left anterior descending coronary, circumflex artery, and heart including pericardium.

As regards the esophagus and main bronchus, a 3 mm planning risk volume (PRV) was also considered.

The clinical target volume (CTV) was identified in accordance with radiation oncology and cardiology, considering the area around PVs, based on anatomical CT imaging, and generating 2 separate target volumes: one around the left PVs (CTVleft) and the other on the right PVs (CTVright).

Based on 4D-CT acquisition, an internal target volume (ITV) was added to CTVs in order to compensate for respiratory motion. Although the CT scan was not synchronized to the electrocardiogram (ECG), the magnitude of ITV was adequate to also include heart displacements [15]. Finally, the planning target volume (PTV) was defined by adding 0–2 mm to the ITV, excluding the overlap area with OaRs/PRV, where PTV was cropped (Figure 1 and Figure 2).

Medial-lateral (M-L), anterior-posterior (A-P) and superior-inferior (S-I) displacements for the center of mass of CTVright and CTVleft were evaluated during all respiratory phases on 4D-CT. STAR was performed in free-breathing, prescribing 25 Gy in single fraction. A “simultaneous integrated protection” (Figure 2) dose was realized to the interface between PTV-PRV to ensure the tolerability of critical structures [16]. To avoid dose hotspots in the ITV-PTV expansion, it was decided to prescribe the dose to ITV volume, with an additional PTV dose coverage of 95% dose to 95% of the volume.

A flattening filter-free (FFF), volumetric-modulated arc therapy (VMAT) plan was generated, normalizing 100% Dp to 95% of the volume, while large intra-target dose heterogeneity D2% (PTV) < 150%Dp was accepted. The radiation treatment was delivered on a linear accelerator (TrueBeam^TM^, Varian Medical System, Mountain View, US). Image-guided radiotherapy (IGRT) using cone-beam CT (CBCT) and surface-guided radiotherapy (SGRT) (Align-RT, Vision RT) were used to reduce setup error and monitor patients during treatment.

In addition, a real time ECG was acquired for the entire procedure for each patient.

The plan conformity index (defined as the volume of 100% of the prescription dose to the volume of PTV) and the gradient measures (GM) were evaluated for all plans.

## 3. Results

From May 2021 to March 2022, 10 elderly patients were treated; the median age was 79.5 years old (range 72–90), and 7 out 10 (70%) patients were female.

In terms of 4D-CT data, the average Cohort displacements of the CTV during the respiratory phases are shown in Figure 3. The average directions for CTVright and CTVleft were −0.01; 0.01; 0.09; 0.01; −0.01; and 0.09 cm, respectively.

However, only S-I movements reported a motion amplitude of 0.6 cm; for M-L and A-P, an amplitude of 0.1 cm was documented.

The main STAR data are summarized in Table 1: average CTVs (right plus left), ITVs (right plus left), and PTVs (right plus left) were 23.6 cc, 44.32 cc, and 62.9 cc, respectively, while the mean prescription isodose level and D2% were 76.5% and 31.2 Gy, respectively.

The plan conformity index and the GM were evaluated for all plans (Table 1), reporting a median value of 1.15 (range 1–2.8), and 1.34, respectively.

In terms of beam arrangement, all of the plans were delivered with three coplanar or non-coplanar 10 MV-FFF arcs, as showed in Table 2.

The mean value of monitor units was 7700 (range), while the mean treatment time was 3 min.

Table 3 reports the dose values observed for all of the OaRs case by case. The average mean dose for heart and LAD were 3.9 and 6.3 Gy, respectively; the mean values observed for the maximum dose of LAD, spinal cord, left and right bronchus, and esophagus were 11.2, 7.5, 14.3, 12.4, and 13.6 Gy, respectively.

Table 4 reports the evaluation of radiation dosage to normal tissue (defined as the total body): Dmean, V5 Gy, and V10 Gy were 0.7 Gy; 3.6%, and 0.8%, respectively.

As the first aim of the trial was to prevent grade 3 toxicities, PTVs were cropped to the PRV of OaRs (Figure 2), so in this way, all dose constraints, in terms of maximal dose for OaRs, were respected [15,17].

## 4. Discussion

In this report, planning and dosimetric analysis on the first 10 patients enrolled in the STAR trial were outlined, with the aim of defining a technical guideline for improving STAR approaches in radiotherapy departments.

In terms of planning parameters, we observed that the majority of patients were planned using three arcs with the combination of 250–110, 50–250, and 320–110 degrees, and 75–35–335 collimator degree. The use of 10FFF was preferred due to the reduction in time of beam exposure, in patients without cardiac devices.

Since there are no other analyses in the literature about the use of LINAC for AF-STAR, we could only compare these results with a few published cases performed with a different technology (Cyberknife) [18,19]. As previously reported, the target definition was not the same. In the Cyberknife cases, the PVs and the left atrial posterior wall were irradiated. The mainstay AF ablation approach is PVs isolation, while appropriate/effective ablation targets, including the atrial wall, remain poorly defined [1,18,19]. Thus, in the present trial, the target was defined as the area around the PVs, defined by radiation oncology and cardiology (Figure 1 and Figure 2). In fact, the present mean CTV was 23 cc, while for Cyberknife, data is roundly 50 cc for all cases.

For all patients enrolled in our phase II trial, a 4D-CT was performed in order to evaluate respiratory movements. CTVLeft and CTVRight displace laterally, A-P and S-I by a few millimeters on average (max amplitude 0.1–0.6 cm). The displacements of CTVLeft and CTVRight were similar during respiratory phases, even if it seems that CTVright was more mobile, with respect to the left, for M-L and A-P displacements (Figure 3).

Based on these results, deep inspiration breath hold (DIBH) could be of limited use for elderly patients with AF (increasing treatment time and difficulty for elderly patients, DIBH could be a trigger of AF), and a free-breathing STAR with 4D-CT should be more efficient.

Based on the targets (2 PTVs for each patient, Figure 1), a 10MV-FFF VMAT plan with three coplanar or non-coplanar arcs is useful in order to obtain better target coverage, sparing OaRs, as shown in Table 2 and Table 3. Surely, considering the “protection of OaRs” and the dose prescription on ITV, the conformity index calculated on PTV was higher than 1, but acceptable (range 1–1.99).

Moreover, Cox et al. showed that, in order to minimize the risk of esophagitis above grade 3, esophagus D2.5 cc and V12 Gy should be less than 14 Gy and 3.78 cc respectively, with a maximum esophagus dose of 22 Gy [20]. For all patients enrolled in the present phase II study, the latter dose constraints for the esophagus were considered and respected.

The OaRs dose differences from LINAC and Cyberknife are negligible: 14 Gy versus 16 Gy for the esophagus, respectively; 4 Gy versus 7.8 Gy for the heart (ventricles) in a Cyberknife case, and 10 Gy (myocardium) in the other two Cyberknife cases [19,21].

The most important advantage of LINAC-based STAR is the delivery time and the MU, and the present report is in line with previous publication reports [3,4,19,21]: treatment time of 3 versus 90 min and MUs of 7700 versus 46,000. However, the shorter time is essential for reducing intrafraction motion, so in the present trial, due to the motion study and IGRT/SGRT monitoring, the introduction of fiducials was not necessary [3,4,5,22].

One limitation that we must mention is, as explained earlier in the text, the lack of synchronization of the CT scan with the ECG; to fill this gap, the magnitude of the ITV was adjusted to effectively include cardiac displacements. Finally, a criticism for STAR in a non-oncological population could be the hypothetical role of low-dose exposure for RT-related carcinogenesis on surrounding healthy tissues. The hypothesis of RT-carcinogenesis derived from atomic bomb survivors, reporting a second tumor probability of 8% at 30 years from primary cancer, with a risk mortality attributable to RT-induced secondary cancer estimated at 1–2% after 10-years [23,24,25,26,27].

Thus, for this reason, the importance of RT-related tumor growth in the elderly population with AF is still negligible. Moreover, in the present analysis, all the conceived dose-volume parameters for the normal tissue structures were low and acceptable (see Table 4: Dmean for all body is 0.7 Gy).

## 5. Conclusions

LINAC-based STAR can be a promising treatment option for elderly patients affected by AF [28], representing a valid non-invasive alternative for elderly patients who were excluded from CA.

The present dosimetric data showed an optimal target coverage, sparing surrounding tissue, in a reduced treatment time, with comfortable results for elderly AF patients. Considering the large diffusion of LINAC in the world, and the large AF population that could be treated with radiation, the present collected data (regarding target definition, motion evaluation, planning, and dosimetry) are interesting for all radiation oncologists who want to implement LINAC-based STAR for AF in their oncological department.

## Figures and Tables

**Figure 1 jpm-13-00596-f001:**
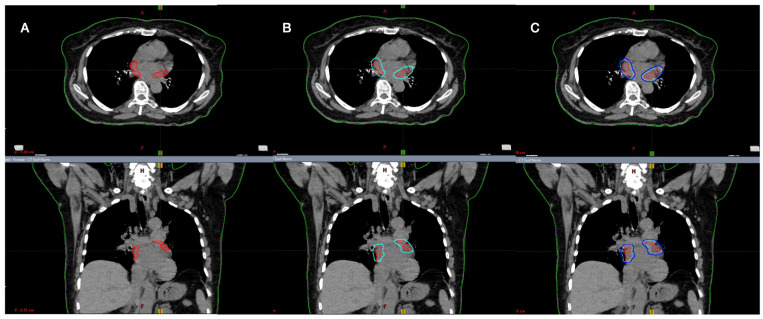
Definition of (**A**) clinical target volume (CTV); (**B**) internal target volume (ITV); (**C**) planning target volume (PTV).

**Figure 2 jpm-13-00596-f002:**
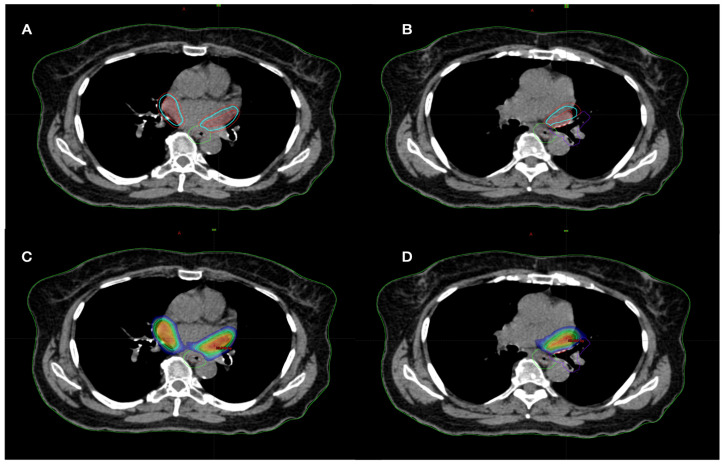
(**A**,**B**) PTV and ITV cropped from OaRs (esophagus and bronchus); (**C**,**D**) STAR Planning with simultaneous integrated protection dose.

**Figure 3 jpm-13-00596-f003:**
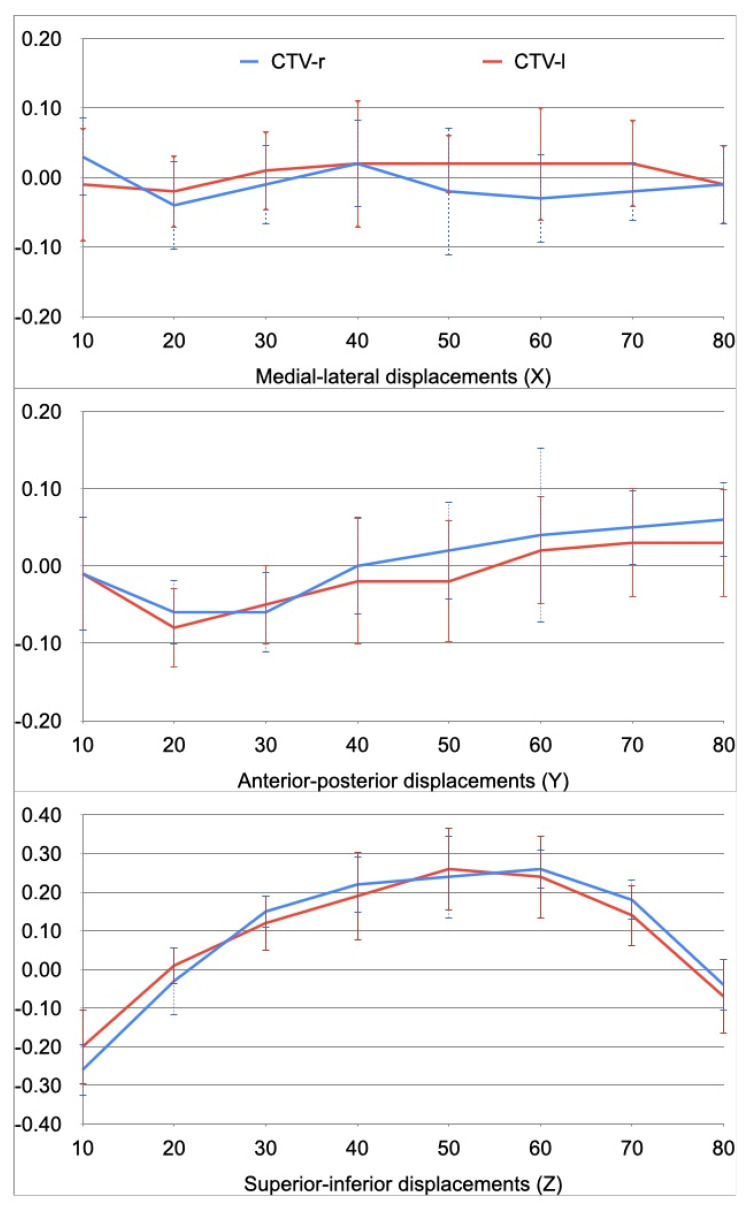
Displacements of CTVs (right CTV-r and left CTV-l) and standard deviation.

**Table 1 jpm-13-00596-t001:** Main treatment planning and dosimetric data.

	PT 1	PT 2	PT 3	PT 4	PT 5	PT 6	PT 7	PT 8	PT 9	PT 10
Age (years)	72	78	84	90	80	86	75	83	77	79
Sex	F	F	F	M	F	M	F	F	M	F
CTV tot (right plus left) (cc)	25.31	12.4	16.3	15.86	25.12	15.83	28.95	18.68	48.05	29.81
ITV tot (cc)	59.10	37.3	38.7	44.55	50.56	32.58	36.57	31.26	68.09	44.65
ITV tot cropped (cc)	54.10	35.2	36	40.22	39.7	28.21	29.19	25.58	63.9	39.13
PTV tot (right plus left) (cc)	78.24	55	58.9	56.6	90.69	49	48.2	43.11	90.15	59.46
PTV tot cropped (cc)	69.88	47.3	53.5	51.71	71.89	42.3	43.8	38	89.6	56.8
Dose 2% (Gy)	31.45	30.6	30.3	30.3	30.2	31	32.6	31.4	33	33
CI	1.80	1.06	1.99	2.82	1.14	1.15	1.09	1.07	1.01	1.00
GM (cm)	1.34	1.28	1.28	1.44	1.60	1.24	1.28	1.31	1.45	1.35
OTT (minutes)	3	3	3	3	3	3	3	3	3	3

F—female; M—male; Gy—Gray; CTV—clinical target volume; ITV—internal target volume; PTV—planning target volume; tot—total (right plus left); OTT—overall treatment time.

**Table 2 jpm-13-00596-t002:** Geometrical characteristics of STAR treatment plans.

Plan #	Collimator Degree	Beam Energy (MeV)	Arcs (Number and Degree)	Couch Angle (Degree)	Monitor Units
#1	25–335–105	10 × FFF	3 arcs 240–110	0	8559
#2	75–35–335	10 × FFF	3 arcs 250–110 50–250 320–110	0–7–353	7622
#3	75–35–335	10 × FFF	3 arcs 250–110 50–250 320–110	0–7–353	7028
#4	75–35–335	10 × FFF	3 arcs 250–110 50–250 320–110	0–7–353	7990
#5	75–35–335	10 × FFF	3 arcs 250–110 50–250 320–110	0–7–353	8014
#6	75–35–335	10 × FFF	3 arcs 250–110 50–250 320–110	0–7–353	8100
#7	75–35–335	10 × FFF	3 arcs 250–110 50–250 320–110	0–7–353	8500
#8	75–35–335	10 × FFF	3 arcs 250–110 50–250 320–110	0–7–353	8040
#9	75–35–335	10 × FFF	3 arcs 250–110 50–250 320–110	0–7–353	6679
#10	75–35–335	10 × FFF	3 arcs 180–179 179–330 30–180	0–5–355	6679

MeV—megaelectron volt; FFF—flattening filter-free.

**Table 3 jpm-13-00596-t003:** Dose constraints of Organs at Risk.

Plan #	Heart Dmean	LAD Dmean	LAD Dmax	Spinal Cord Dmax	Left Bronchus Dmax	Right Bronchus Dmax	Esophagus Dmax	Esophagus D2.5	Esophagus V12
#1	4.7 Gy	4.5 Gy	6.5 Gy	4 Gy	10.8 Gy	4.6 Gy	11.8 Gy	9.37 Gy	/
#2	3.9 Gy	6.5 Gy	12 Gy	8.3 Gy	8.1 Gy	18 Gy	10.7 Gy	8.59 Gy	/
#3	3.8 Gy	7.8 Gy	13 Gy	7.1 Gy	15 Gy	14 Gy	12.9 Gy	10.96 Gy	0.25 cc
#4	4 Gy	6.3 Gy	9 Gy	7.1 Gy	19 Gy	6.7 Gy	13.2 Gy	9.38 Gy	0.12 cc
#5	4.2 Gy	4.1 Gy	11.2 Gy	8.9 Gy	19 Gy	16.4 Gy	14.1 Gy	11.4 Gy	0.91 cc
#6	3.7 Gy	8 Gy	18 Gy	7.7 Gy	10 Gy	7.5 Gy	15.4 Gy	11.48 Gy	1.38 cc
#7	3.4 Gy	8.9 Gy	14.8 Gy	7.7 Gy	15.7 Gy	16.1 Gy	14.2 Gy	11.46 Gy	0.94 cc
#8	3.84 Gy	3.9 Gy	8.3 Gy	6 Gy	19 Gy	12.6 Gy	15.2 Gy	12.07 Gy	2.76 cc
#9	3.7 Gy	8 Gy	11 Gy	10.4 Gy	15 Gy	15 Gy	14.6 Gy	11.03 Gy	0.91 cc
#10	4.4 Gy	5 Gy	8.6 Gy	8.4 Gy	11.2 Gy	13 Gy	14.7 Gy	11.74 Gy	1.55 cc

Dmean—mean dose; Dmax—maximum dose; Heart—heart minus PTV (planning target volume); LAD—left anterior descending artery; Gy—Gray; D2.5—dose to 2.5 cc; V12—volume receiving 12 Gy.

**Table 4 jpm-13-00596-t004:** Dosimetric parameters for normal tissue.

Plan #	Dmean (Gy)	V5 (%)	V10 (%)
#1	0.7	3.9	0.85
#2	0.68	4	0.9
#3	0.58	3	0.7
#4	0.7	3.9	0.8
#5	0.9	5.8	1.3
#6	0.7	3.4	0.8
#7	0.65	3	0.7
#8	0.65	2.9	0.7
#9	0.5	2.7	0.6
#10	0.6	3.7	0.9

Dmean—mean dose; Gy—Gray; V5—volume receiving 5 Gy; V10—volume receiving 10 Gy.

## Data Availability

The data can be acquired from the corresponding author.

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
