# Peer review of "Phase II Trial of LINAC-Based STereotactic Arrhythmia Radioablation (STAR) for Paroxysmal Atrial Fibrillation in Elderly: Planning and Dosimetric Point of View"

_jpm, 2023, doi:10.3390/jpm13040596_

Round 1

Reviewer 1 Report

From the cardiologist point of view, this is a well written manuscript regarding a novel indication of star, which, in the future, could become very important to cure AF, although data is missing concerning the target population for this technique.

There are two issues in the Methods section, page 5,  that need to be clariried:

-CT scan was not synchronized to the ECG. This is a major problem when we are talking about a moving target. Of course I understand authors did not have an alternative but they need to mention this is the limitations section, which does not exist so far in this paper as a separate section.

-On the other hand authors use the achronims OaRs/PRV without previous decription. Please fix this.

Finally I don´t feel qualified to judge the radiooncologist part of this manuscript.

Author Response

Point 1. CT scan was not synchronized to the ECG. This is a major problem when we are talking about a moving target. Of course I understand authors did not have an alternative but they need to mention this is the limitations section, which does not exist so far in this paper as a separate section.

Response 1: Thank you very much for your manuscript revision and for your suggestions.

We added in the text a “limitation section” in which we better explained the concept.

This sentence was added: “One limitation that we must mention is, as explained earlier in the text, the lack of synchronization of the CT scan with the electrocardiogram; to fill this gap, the magnitude of the ITV was adjusted to effectively include cardiac displacements.”

Point 2: On the other hand authors use the achronims OaRs/PRV without previous decription. Please fix this..

Response 2: Thank you, we added the explanation of these achronims “organs at risks (OaRs) and planning risk volume (PTV)”.

Reviewer 2 Report

Dear authors, I have read with great pleasure the manuscript which I consider excellent for any aspect of the procedural workflow (pre-treatment imaging, movement management, treatment planning, quality assurance, treatment delivery).

Author Response

Point 1. Dear authors, I have read with great pleasure the manuscript which I consider excellent for any aspect of the procedural workflow (pre-treatment imaging, movement management, treatment planning, quality assurance, treatment delivery).

Response 1: Thank you for your revision and for your words of appreciation for our manuscript.

Reviewer 3 Report

Dear Editor and Authors,

Thank you for asking me to review this work titled “Phase II trial of LINAC-based STereotactic Arrhythmia Radioablation (STAR) for paroxysmal atrial fibrillation in elderly: planning and dosimetric point of view” by Dr. Bonaparte from the Department of Radiation Oncology at the General Regional Hospital "F. Miulli" in Bari, Italy.

In this a quite interesting small prospective phase II/pilot study the authors have utilized stereotactic arrhythmia radioablation (STAR) to treat 10 patients with atrial fibrillation. The authors herewith report their protocol and their safety and dosimetric outcomes of the delivered treatment. Their radiotherapy protocol seems correct, although I must confess it is outside my everyday practice.

The manuscript is well written, clear to understand and well presented and illustrated/supported with figures and tables. The language only requires some minor grammatical/expression corrections by a native speaker.

I only have some very few comments to make:

1.    Why did the authors not report additional demographic information such as duration of AF, therapies used/medication, ect?

2.    Some of the results/outcomes reported in the discussion (for example grade III toxicity) should have been reported in the result section!

In conclusion, this is an interesting report and I am keen to support its presentation following some very minor edits suggested.

Thank you,

Author Response

Point 1. Why did the authors not report additional demographic information such as duration of AF, therapies used/medication, ect?

Response 1: Thank you for your revision and for your questions.

We think that reporting further demographic or specific information relating to the therapies used for AF is not in line with the objective of the manuscript which focuses, in fact, on the evaluation of alternative treatments to the already known and widely used standard procedures. Of course, if the author deems it necessary, we could include these specifications in the text.

Point 2. Some of the results/outcomes reported in the discussion (for example grade III toxicity) should have been reported in the result section!

Response 2: Thank you for your suggestion. We modified the text by moving the part relating to toxicity to the "results" section. This is the moved sentence: “As the first aim of the trial was to prevent grade 3 toxicities, PTVs were cropped to PRV of OaRs (Figure 2), so in this way, all dose constraint, in terms of maximal dose, for OaRs were respected.”

We modified all references accordingly.